# IIB–CPE: Inter and Intra Block Processing-Based Compressible Perceptual Encryption Method for Privacy-Preserving Deep Learning [note 1]

**DOI:** 10.3390/s22208074

**Published:** 2022-10-21

**Authors:** Ijaz Ahmad, Seokjoo Shin

**Affiliations:** Department of Computer Engineering, Chosun University, Gwangju 61452, Korea

**Keywords:** image coding, JPEG compression, perceptual encryption, privacy-preserving deep learning

## Abstract

Perceptual encryption (PE) of images protects visual information while retaining the intrinsic properties necessary to enable computation in the encryption domain. Block–based PE produces JPEG-compliant images with almost the same compression savings as that of the plain images. The methods represent an input color image as a pseudo grayscale image to benefit from a smaller block size. However, such representation degrades image quality and compression savings, and removes color information, which limits their applications. To solve these limitations, we proposed inter and intra block processing for compressible PE methods (IIB–CPE). The method represents an input as a color image and performs block-level inter processing and sub-block-level intra processing on it. The intra block processing results in an inside–out geometric transformation that disrupts the symmetry of an entire block thus achieves visual encryption of local details while preserving the global contents of an image. The intra block-level processing allows the use of a smaller block size, which improves encryption efficiency without compromising compression performance. Our analyses showed that IIB–CPE offers 15% bitrate savings with better image quality than the existing PE methods. In addition, we extended the scope of applications of the proposed IIB–CPE to the privacy-preserving deep learning (PPDL) domain.

## 1. Introduction

The recent surge in cloud-based computing services, the popularity of social networking services, and cloud-based storage have motivated the exchange of a large amount of information over the Internet. One of the most popular applications is the computation of deep learning (DL) models. However, data transmission over public networks is always at risk of information leakage, and a large volume of data requires large bandwidth. In addition, computation on privacy-sensitive data should preserve the data confidentiality. One solution is to encrypt and compress datasets. Traditional number theory and chaos theory-based full encryption algorithms [1,2,3,4] have proven to be efficient in protecting multimedia data confidentiality during transmission; however, they do not support privacy-preserving computation. Therefore, these methods are not suitable for outsourcing privacy sensitive data to cloud servers. In recent years, perceptual encryption (PE) of images has emerged as an alternative that provides the necessary level of security while allowing computation over encrypted data. PE methods can be classified as learnable or compressible encryption methods. Learnable PE performs pixel-level image obfuscation and is designed for PPDL [5,6,7,8]. However, the methods described in [5,6] are vulnerable to chosen-plaintext attacks, as demonstrated by [8,9], and the method described in [7] has resulted in poor performance of DL models. On the other hand, compressible PE are block-based methods that produce compressible images called encryption-then-compression (EtC) [10,11,12,13,14,15,16]. The applications of EtC schemes have been extended to traditional machine learning (ML) techniques such as support vector machines (SVM) for privacy-preserving face recognition tasks in [17,18]. However, in [17], the input images are grayscale with one channel, thereby omitting the step necessary to expand the keyspace proposed in [13], whereas, the work in [18] features a format compatibility issue. In [19], the authors propose an isotropic network that uses EtC images to enable privacy-preserving classification. To match the patch size of their classification model, they used a larger block size of (16 × 16) during encryption. However, the use of such a larger block size may raise security concerns, as discussed in [20]. In addition, existing PE algorithms do not consider PPDL systems in a holistic manner, that is, they do not consider encryption performance, compression savings, and DL model accuracy.

The encryption algorithm of the EtC system is block-based and performs four steps: *block permutation, block rotation, block inversion, and negative–positive transformation*. The steps are computationally inexpensive and encrypt images in such a way that resulting cipher images retain their intrinsic properties necessary for compression. Nonetheless, EtC schemes are robust against various types of attacks, including brute-force and cipher-text-only attacks [20]. These schemes have the additional advantage of being JPEG-compatible, which makes them suitable for several applications, such as cloud-based photo storage and social networking services [13,14], image retrieval systems in the privacy domain [21], and medical image services [12,22]. EtC schemes have two categories based on the input image representation. First, color–EtC schemes [10,11], where the input is represented as a three-channel image, have an additional step of block shuffling among the color channels. Second, grayscale-EtC schemes (GS–EtC) [13,14], where the input is represented as a pseudo-grayscale image by combining the color channels along the horizontal or vertical direction. GS–EtC algorithms have better security than color–EtC schemes because they can use a smaller block size. However, the lack of color information limits their applications, and they cannot be applied to PPDL.

In this study, we first extended the applications of color–EtC schemes to the PPDL domain. Given their limitations, we proposed a method to improve the security efficiency without compromising compression savings and DL model accuracy. The proposed method is called inter and intra block-based compressible perceptual encryption (IIB–CPE). IIB–CPE performs the encryption steps that only change correlation’s direction on a sub-block level, thereby improving the encryption efficiency. In other words, if the orientation of a sub-block in a block is changed, it is difficult to recover the correct orientation of the entire block without reconstructing the entire image [23]. The preliminary results of this study are presented in [12], where the method was applied for grayscale image compression and encryption. In particular, the main purpose was to provide security and bandwidth efficiency during transmission only.

The main contributions of this work are summarized as follows: (1) An efficient block-based compressible perceptual encryption algorithm is proposed, which eliminates the security vulnerabilities of existing PE methods. The proposed scheme uses inter and intra block processing, which allows a smaller block size, thus expanding the keyspace of the encryption algorithms and providing resistance against different attacks. (2) The proposed method can provide security and bandwidth efficiency during image data transmission, and can enable privacy-preserving computation on third-party cloud resources without disclosing the information. (3) For the first time, the current study explained why PE cipher images are compressible, to the best of our knowledge. (4) The study proposed an extended jigsaw puzzle solver to accommodate the sub-block-level processing. (5) The experiments were conducted on three different real image datasets to demonstrate the encryption efficiency, compression savings, and PPDL-based classification.

The rest of the paper is organized as: Section 2 summarizes related work of PE methods and their applications in PPDL. Section 3 presents conventional PE methods and proposed IIB–CPE method and its principal design. Comparison between existing and proposed PE methods in terms of compression savings, encryption efficiency, and PPDL performance is drawn in Section 4 and Section 5. Finally, the paper is concluded in Section 6 with notes on the future research direction.

## 2. Related Work

### 2.1. Perceptual Encryption Methods

The conventional encryption algorithms are applicable for encrypting uncompressed raw images as they always perform stream encryption and/or scrambling of pixels values. However, they are not suitable for encrypting the JPEG compressed images as they may disturb the markers and render them uninterpretable. In addition, re-encoding the encrypted image as a JPEG image may increase the file size. Therefore, encryption of JPEG compressed images has some additional requirements compared to uncompressed images: (1) the encrypted image should be JPEG format-compatible, (2) there should be no or small increment in the encrypted image size, and (3) the encryption should provide a necessary level of security. Overall, an encryption algorithm should provide a balance between security and usability.

To meet these requirements, perceptual encryption algorithms have been proposed, which reverse the conventional order of performing compression prior to encryption and they are called encryption-then-compression (EtC) methods. It may seem inefficient in a sense that the encryption process may have destroyed the correlation present in a plain image that is exploited by a compression algorithm to provide reduction in image size. However, the EtC encryption algorithm protects only perceptual information of an image and preserves its intrinsic properties necessary for compression.

A block-based EtC scheme compatible with the JPEG standard is proposed in [10,11] and lossless image compression standards are proposed in [15]. However, the use of a larger block size and the same key for each color component may make the scheme vulnerable to jigsaw puzzle attack. In [16], the authors proposed to perform block rotation, inversion, and negative-transformation steps independently in each RGB component. The independent processing of each color component increased the number of blocks for better security than achieved in [10,11,15]. However, this resulted in a format compatibility issue, that is, the methods are only applicable with the JPEG lossless algorithm. The compatibility issue of [16] is addressed in [18] where orientation and pixel values in a block are changed independently in each color component. The method shuffled the blocks’ positions in each component with the same key, which allows the use of YCbCr color space in the JPEG algorithm for improved compression savings. The extended EtC methods proposed in [16,18] provide better security as the independent processing of color components expanded the keyspace and altered the color distribution significantly. However, their main limitation is that they cannot use a smaller block size for better security. The larger block size has been used to avoid color distortion in the recovered image when the JPEG compression is performed with chroma subsampling.

An alternative approach is adopted in [13] to allow the use of a smaller block size. The method represents an input color image as a pseudo grayscale image by concatenating the color components along the horizontal or vertical direction. The grayscale representation allows the use of a smaller block size without any adverse effect on the recovered image quality. In addition to smaller block size, lack of color information improves robustness of the method against jigsaw puzzle attacks. However, such representation does not allow the JPEG chroma subsampling. In [14], the authors propose a somewhat similar method to the one in [13] with the difference that an input RGB image is first converted to YCbCr color space and performed chroma subsampling before representing the image as a pseudo-grayscale image. Overall, for EtC schemes, there is an efficiency tradeoff between compression and encryption because of the block size. In addition, existing EtC methods have a prerequisite of color image as an input for better security. For example, the extended EtC methods process each color component independently to achieve a larger number of blocks. Similarly, GS–EtC methods represent an input color image as a pseudo grayscale image to benefit from the JPEG smaller block size to achieve a large number of blocks. However, in applications such as ones in medical image processing domain, the unavailability of color images makes the methods obsolete. In order to resolve these limitations, this work proposed inter and intra block processing for compressible perceptual encryption methods (IIB–CPE). In addition to digital image encryption, optical encryption methods have been proposed to process large amount of information efficiently [24] and their cipher images compressibility was studied in [25].

### 2.2. Privacy-Preserving Deep Learning Methods

Compare to full-image encryption methods, the PE algorithms make the images visually unrecognizable by hiding only the perceptual information in them. This information can be exploited to enable different image processing and computer vision algorithms computations in encryption domain. For example, [17,18] extended applications of block-based PE algorithms to SVM-based privacy-preserving face recognition for grayscale and color images, respectively. In [19], the authors propose an isotropic network that uses color–EtC images for PPDL. The main advantage of their method is that it does not require an adaptation network to achieve classification accuracy on the encrypted images comparable to that of the plain images. On the other hand, learnable PE methods have been proposed to enable PPDL applications in [5,6,7,8,26,27,28]. In [5], they have proposed a PE method that forms a 6-channel image from an RGB image by splitting predefined blocks into upper and lower 4-bit images. Then, the encryption is achieved by randomly changing pixels intensities and shuffle their positions. The method has been successfully applied to PPDL; however, it is vulnerable to chosen plaintext attack demonstrated in [9]. To deal with the security vulnerability of [5], an extended version of their method is proposed in [7], which uses a different key in each channel. However, it has severely degraded the DL model performance. In [6], they have proposed to randomly change pixels intensities and shuffle the color components in an image. The method is applicable to PPDL; however, it is vulnerable to chosen plaintext attack, as demonstrated in [9]. The main reason for the scheme vulnerability is the use of a single value subtraction from the pixel intensities. Therefore, the authors of [8] proposed to xor a random sequence generated by chaotic map with the pixel values for better security without compromising the DL performance. However, when using chaos theory-based image encryption algorithms, careful consideration is required to avoid weak and equivalent keys in order to resist different types of attacks [4]. Alternatively, [26] proposed to divide the encrypted image of [6] into blocks and apply three different types of filters on randomly selected blocks. Furthermore, the aforementioned methods that perform digital encryption, [27,28], have proposed optical image encryption methods to take advantage of parallel computing for PPDL.

## 3. Methods

The compressible PE methods mainly consists of the following steps:

Step 1. Input image representation: When an input is a color image, it can either be represented as a true color image with three-color components or as a pseudo-grayscale image with only one component. The pseudo-grayscale image can be obtained by concatenating the color components in either horizontal or vertical direction.

Step 2. Block-based encryption that performs geometric and color transformations. A geometric transformation is performed to change positions and orientations of the blocks, whereas color transformation is performed to modify pixels intensities.

Step 3. The encrypted image still preserves certain properties of the original image on a block level that can be exploited for compression. Specifically, when the block size is maintained at either (16 × 16) or (8 × 8) during encryption, then the cipher image can be compressed with the JPEG standard in either color or grayscale mode, respectively.

Based on these three steps, several CPE methods have been proposed as discussed in Section 2. The following subsections describe existing PE methods and proposed PE method with its principal design.

### 3.1. Conventional Compressible Perceptual Encryption Methods

Step 1. Divide the input image$\;I_{RGB}\;$with W×H×C pixels into non-overlapping blocks, each with$\;B_{M}\times B_{N}\;$pixels, and permute their locations using a randomly generated secret key$\;K_{1}$.

Step 2. Apply rotation–inversion randomly to each block using a key K2 where each entry represents rotation degrees and flipping axis.

Step 3. Apply a negative–positive transformation to each pixel in a block randomly chosen by a uniformly distributed binary key$\;K_{3}$. The modified value $\;p^{\prime }\left(j,k\right)\;$ where  0≤j≤M, 0≤k≤N  of a pixel  p(j,k)  in the original image is:(1)$$p^{\prime }\left(j,k\right)=\{\begin{array}{cc} \;\;\;\;\;p\left(j,k\right), & K_{3}\left(i\right)=0\\ 255-p\left(j,k\right), & K_{3}\left(i\right)=1 \end{array}$$

Step 4. Shuffle the blocks among the color channels using key$\;K_{4}\;$where each entry represents a unique permutation of the color components.

Step 5. Finally, apply JPEG compression to the cipher image.

The above steps provide a general framework for the conventional PE methods, which can be used to describe each conventional EtC method discussed in Section 2.1. For example, when the input image I is represented as a color image with three components such as red (*R*), green (*G*), and blue (*B*), then the key used in each encryption step i∈{1, 2, 3, 4}, is$\;K_{i}\in \left\{K_{i}^{R},\;K_{i}^{G},\;K_{i}^{B}\right\}$. The color–EtC methods proposed in [10,11,15] use the same key in each color component for encryption steps i∈{1, 2, 3} as$\;K_{i}^{R}=K_{i}^{G}=K_{i}^{B}$. The use of same key makes the systems vulnerable to jigsaw puzzle attack. The extended EtC method [16] proposed to process each color component independently by using different keys as$\;K_{i}^{R}\neq K_{i}^{G}\neq K_{i}^{B}\;$in steps i∈{1, 2, 3} for improved security. However, this results in a format compatibility issue, which has been addressed in [18]. The method proposed to change blocks orientation and perform negative–positive transformation independently in each color component as$\;K_{i}^{R}\neq K_{i}^{G}\neq K_{i}^{B}\;$for i∈{2, 3}, while shuffling their positions with the same key. The extended EtC methods dealt with the limited keyspace of color–EtC methods; however, their main limitation is that a block size of no less than (16 × 16) should be used. On the other hand, the input image can be represented as a pseudo grayscale image by concatenating the color components along the horizontal or vertical direction, as proposed in [13,14]. This representation allows to use a smaller block size of (8 × 8) as the cipher image can be compressed by the JPEG algorithm in grayscale mode. The extended EtC methods and GS–EtC methods have improved encryption efficiency as they have significantly altered the color distribution and expanded the keyspace. Nonetheless, the key size expansion is limited by the smallest allowable block size used in the JPEG compression algorithm and the lack of color information limits their applications.

#### Conventional EtC Methods for Grayscale Image Encryption

Different from color images, a grayscale image consists of only one channel. Therefore, the conventional EtC methods can be extended for grayscale image encryption simply by omitting Step 4 in Section 3.1. The EtC method for grayscale image encryption is described below:

Step 1. Divide the input grayscale image $\;I_{GS}\;$ with  W×H  pixels into non-overlapping blocks, each with $\;B_{M}\times B_{N}\;$ pixels, and permute their locations using a randomly generated secret key$\;K_{1}$.

Step 2. Apply rotation–inversion randomly to each block using a key K2 where each entry represents rotation degrees and flipping axis.

Step 3. Apply a negative–positive transformation to each pixel in a block randomly chosen by a uniformly distributed binary key $\;K_{3}$ as in Equation (1).

Step 4. Finally, apply JPEG compression in grayscale mode to the cipher image.

For grayscale image encryption, the conventional EtC methods (Color–EtC, Extended EtC and GS–EtC methods) all can be implemented with the same steps described above. More importantly, the keyspace expansion resulted from independent processing of color components (in the case of Extended EtC) and pseudo-grayscale image generation to benefit from smaller block size (in the case of GS–EtC) become obsolete. It is worth mentioning that the JPEG compression is performed in grayscale mode; therefore, the smallest allowable block size (that is 8 × 8) can be used during encryption.

### 3.2. Proposed Method

#### 3.2.1. Principle Design: Inside–Out Transformation

In natural images, on average, eight pixels are spatially correlated in either direction; therefore, in the JPEG compression algorithm a block size of (8 × 8) is used for better compression, and for manageable computational complexity and memory requirements. For compressibility of the perceptually encrypted images, neighboring pixels must have the same correlation as the original images. Therefore, in conventional methods, encryption steps are inter block processes that transform an entire block to preserve the intrinsic properties of an image as shown in Figure 1. However, different steps of the scheme affect the correlation differently. For example, block permutation and negative–positive transformation steps change the correlation coefficient, whereas rotation and inversion only change the correlation direction. Therefore, performing the later steps at a sub-block level (intra block process) may preserve compression savings while improving encryption efficiency. In addition, the JPEG color sampling artifacts can be avoided since the correlation is preserved within a block. The basic principle is to divide blocks into sub-blocks and implement scrambling inside each block to achieve visual encryption of local details. The goal is to preserve global contents such as spatial information and correlation among adjacent pixels, which can be used to enable several applications in encryption domain. The intra block processing results in an *inside–out transformation* that disrupts symmetry of an entire block as opposed to inter block processing of conventional methods. The geometric transformations performed only on an entire block are rigid motions that preserve symmetry of a block, that is, pixels on edges remain the same. As discussed in Section 2, existing PE methods have a prerequisite of color image as an input for better security. As opposed to them, the encryption efficiency of a proposed method is independent of input image representation because of the intra block processing. Therefore, sub-block operations make the proposed method suitable for both grayscale and color images.

#### 3.2.2. Proposed Compressible Perceptual Encryption Method

Figure 1 shows a high-level comparison between the conventional PE and proposed IIB–CPE methods. The proposed algorithm performs block-level inter processing and sub-block-level intra processing, as opposed to the existing EtC methods that transform an entire block. A detailed comparison is performed in Figure 2. The proposed inside–out transformation that shuffles pixels inside a block is shown in Figure 2. IIB–CPE consists of the following block-based steps:

Step 1. Divide the input image $I_{RGB}$with  W×H×C  pixels into non-overlapping blocks, each with $\;B_{M}\times B_{N}\;\;$ pixels, and permute them using a randomly generated secret key$\;K_{1}^{inter}$.

Step 2. Apply a negative–positive transformation to each pixel in a block randomly chosen by a uniformly distributed binary key  K2inter as in Equation (1).

Step 3. Shuffle the blocks among the color channels using key$\;K_{3}^{inter}$ where each entry represents a unique permutation of the color components.

Step 4. Perform inside–out transformation of an entire block as:

Divide each block into sub–blocks of size $\;B_{M_{intra}}\times B_{N_{intra}}\;$ pixels;

Apply rotation–inversion randomly to each sub-block using a key $\;K_{4}^{intra}$ where each entry represents rotation degrees and flipping axis.

Step 5. Finally, apply JPEG compression to the cipher image.

The proposed PE method is symmetric-key algorithm that requires the same set of keys used for both encryption of plain images and decryption of cipher images. Figure 3a shows an example image from Tecnick dataset and its cipher images (b) and (c–e) obtained from Color–EtC and proposed methods, respectively. For visual analysis, bottom left corner in each cipher image is enlarged. It can be seen that proposed method achieves high visual encryption of local details. According to the application requirements, the encryption level can be controlled by performing the above encryption steps only on selected blocks. The shuffling key$\;K_{1}\;$consists of blocks permutations that map each block to a random location in the output image. Scrambling only selected blocks can retain the global information of an image. Similarly, the key $\;K_{i}\;$ where  i∈{2,3,4} of the *i*th step consists of an element (for example, 0) that represents an identity map. Therefore, increasing the number of such element can decrease the level of encryption. In addition, the use of larger sub-blocks can preserve local information inside a block.

#### 3.2.3. IIB–CPE for Grayscale Image Encryption

Similar to the conventional EtC methods, the proposed IIB–CPE can be extended for the grayscale image encryption simply by omitting Step 3 in Section 3.2.2. Figure 4 shows extension of proposed method for grayscale image encryption is described below:

Step 1. Divide the input image $\;I_{GS}\;$ with  W×H×C  pixels into non-overlapping blocks, each with$\;B_{M}\times B_{N}\;$pixels, and permute them using a randomly generated secret key$\;K_{1}^{inter}$.

Step 2. Apply a negative–positive transformation to each pixel in a block randomly chosen by a uniformly distributed binary key $\;K_{2}^{inter}$ as in Equation (1).

Step 3. Perform inside–out transformation of an entire block in the following way:

Divide each block into sub-blocks of size $\;B_{M_{intra}}\times B_{N_{intra}}\;$ pixels;

Apply rotation–inversion randomly to each sub-block using a key $\;K_{3}^{intra}$ where each entry represents rotation degrees and flipping axis.

Step 5. Finally, apply JPEG compression in grayscale mode to the cipher image.

In Step 1 and Step 2, a block size (8 × 8) can be used, while in Step 3, smaller block sizes such as (4 × 4) and (2 × 2) can be used.

## 4. Simulation Results and Analysis

In this section, we first evaluate compression savings and encryption efficiency of the IIB–CPE. For the evaluation, we conducted our experiments on two datasets. The Tecnick sampling dataset [29] comprising 120 true color images with dimensions of 1200 × 1200 and Shenzhen dataset [30] consisting 400 grayscale images with dimensions of 2048 × 2048. To demonstrate the application of the proposed method, we designed an end-to-end image communication system for PPDL-based image classification. For the classification accuracy, we used the CIFAR datasets [31]. Both CIFAR–10 and CIFAR–100 consist of 50K training and 10K testing images. For the baseline methods, we implemented compressible PE algorithms proposed in [10,11] (Color–EtC), [13,14] (GS–EtC), and learnable PE methods, such as learnable encryption (LE) [5], pixel-based PE (PBE) [6], chaos-based secure PE (SPBE) [8], and extended learnable encryption (ELE) [7]. For the JPEG software, we used the implementation provided in [32].

### 4.1. Energy Compaction Analysis

At the core of JPEG is the discrete cosine transform (DCT) [33], which reduces the data correlation and provides a compact representation of a large amount of information as few data samples. The JPEG standard divides an image into $\;N^{2}=8^{2}\;$ blocks of pixels, with each block producing 64 coefficients. The DCT–II for a block, where  I(i,j)  is the pixel value at position  (i,j), is defined as:(2)$$F\left(u,v\right)=\alpha \left(u\right)\alpha \left(v\right)\overset{N-1}{\underset{i=0}{\sum }}\overset{N-1}{\underset{j=0}{\sum }}I\left(i,j\right)\times \cos\left[\frac{\left(2i+1\right)\pi u}{2N}\right]\cos\left[\frac{\left(2j+1\right)\pi v}{2N}\right],$$
where
$$\alpha \left(x\right)=\{\begin{array}{cc} \sqrt{\frac{1}{N}} & x=0\\ \sqrt{\frac{2}{N}} & x>0 \end{array}$$
and F(u,v) is the corresponding computed DCT value. For the DC coefficient, u=v=0 and  F(0,0)=1/N∑∑I(i,j), which gives the average value of pixels in the image block. Therefore, the DC coefficient is independent of the pixel position. The remaining 63 values are AC coefficients that correspond to a progressive increase in frequency in both the horizontal and vertical directions. When the input image is divided into multiple blocks, the DCT can be independently computed for each block. In each calculation, the basis function points remain the same, whereas the pixel values are changed. Therefore, an efficient method is to pre-compute the function points and multiply them by each block to obtain DCT (*D*) as
(3)$$D=TMT^{\prime }$$
where M is the image data and T is the DCT matrix obtained as
$$T_{i,j}=\{\begin{array}{cc} \;\;\;\;\;\;\;\;\;\;\;\;\;\;\frac{1}{\sqrt{N}}, & if\;i=0,\\ \sqrt{\frac{2}{N}}\cos\left[\frac{\left(2j+1\right)i\pi }{2N}\right], & otherwise. \end{array}$$

The multiplication of T on the left transforms the rows, and $T^{\prime }\;$ on the right transforms the columns of M. The product PM represented by P is a linear combination of matrix  T  columns with weights given by matrix  M  columns. The matrix  M  with c columns, each with r elements, can be represented in a column–vector form as $\;M=\left[M_{0},M_{1},\ldots ,M_{c-1}\right]$, where for i=0, 1, ⋯, c−1, $M_{i}\;$ denotes the ith column of M, and for k=0, 1, ⋯, r−1, $m_{ki}\;$denotes kth element of the ith column. The product P is obtained as $\;P=\left[TM_{0},TM_{1},\ldots ,TM_{c-1}\right]$, where the ith column of P is $\;P_{i}=TM_{i}$ and is defined as a product of matrix T with the ith column of M. It is calculated by
(4)$$P_{i}=\overset{r-1}{\underset{k=0}{\sum }}T_{*k}m_{ki}$$

This representation simplifies the change in the product matrix elements with respect to weight. For example, in Equation (4), the weights belong to an image distribution, which has the intrinsic property of being highly correlated; thus, the adjacent pixels have smaller differences. Therefore, swapping the pixel positions in a block by intra block processing has a smaller effect on the product of the two matrices. As an example, we extracted an 8 × 8 block from the Lena image, where the vertical and horizontal correlation factors are$\;\sigma_{v}=0.6\;$ and $\sigma_{h}=0.7$, respectively. The DCT transformations of the original, conventional PE, and IIB–CPE images are shown in Figure 5a,c. The PE image is obtained by 90° rotation of the whole block, whereas the IIB–CPE is obtained by first dividing the block into 4 × 4 sub-blocks and then randomly and independently rotating and inverting each sub-block. Figure 5b illustrates that the transformation of the entire block only changes the DCT coefficient positions, while their values remain the same as those of the original image. On the other hand, the sub-block processing results in DCT coefficients with values slightly different from the plain image, as shown in Figure 5c. However, the transformation in the JPEG algorithm is followed by a quantization step, to reduce each coefficient value obtained in Equation (2) [34]. The reduction amount is controlled by a quantization table  QT, which is defined by a quality factor  QF∈{1,2,⋯,100}. The JPEG standard provides two quantization tables $\;{\mathrm{QT}}_{50}$ for  QF=50%, one for each image component. The standard tables can be used to construct different quantization tables corresponding to different quality factors. The element$\;qt_{\mathrm{QF}}\left(u,v\right)$ of a quantization table  QT for a quality factor qf% is defined as
(5)$$qt_{qf}\left(u,v\right)=\{\begin{array}{cc} \;\;\;\;\;G\left(\lfloor \frac{qt_{50}\left(u,v\right)\times \left(\frac{5000}{qf}\right)+50}{100}\rfloor \right), & qf<50\\ G\left(\lfloor \frac{qt_{50}\left(u,v\right)\times \left(200-2qf\right)+50}{100}\rfloor \right), & qf>50 \end{array}$$
where the function  G(x)  ensures that the elements in Equation (5) remain integers and are between 1 to 255 as required by the standard recommendation. The function is defined as
$$G\left(qt_{qf}\left(u,v\right)\right)=\{\begin{array}{cc} \;\;\;\;\;\;\;1, & qt_{qf}\left(u,v\right)<1,\\ \;\;\;\;\;255, & \;\;qt_{qf}\left(u,v\right)>255,\\ qt_{qf}\left(u,v\right), & \mathrm{otherwise}.\;\;\;\; \end{array}$$

The DCT coefficients computed in Equation (2) can be quantized by dividing each value  F(u,v)  by the corresponding quantization element $\;qt_{\mathrm{QF}}\left(u,v\right)\;$as
(6)$$F^{’\left(u,v\right)}=\lfloor \frac{F\left(u,v\right)}{qt_{\mathrm{QF}}\left(u,v\right)}\rfloor ,$$
where function ⌊⋅⌋rounds a value to the closest smallest integer and $F^{\prime }\left(u,v\right)\;$ is the quantized value. Since the quantization reduces the values to small integers, the difference in the DCT coefficients resulting from the sub–block processing was significantly reduced in the JPEG quantization step, as shown in Figure 5d–f. In fact, the number of zeros at the end of the zigzag scan remains the same and can be encoded as the JPEG end-of-block (EOB) identifier in a manner similar to the plain image. Therefore, the IIB–CPE has little impact on the energy compaction of the DCT and it allows the quantizer to discard high-frequency coefficients without introducing any visual distortion in the recovered image.

### 4.2. Compression Analysis

For compression analysis, we plotted the rate distortion (RD) curve as shown in Figure 6, Figure 7 and Figure 8 for color image compression without chroma subsampling, color image compression with 4:2:0 subsampling ratio, and grayscale images, respectively. The y-axis is the recovered image quality represented as a multiscale structural similarity index measure (MS–SSIM) in dB against the compression savings given as the bitrate on the x–axis. The RD curves are for the JPEG quality factors of 70–100. To compare the RD curves, we used Bjøntegaard delta (BD) measures [35], where the BD rate difference is the percent difference between two bitrates of the equivalent quality, and the BD quality difference is the average dB difference for the equivalent bandwidth. Instead of peak signal-to-noise ratio (PSNR), as used in the conventional literature, the BD rate difference is computed for a better image quality measure such as MS–SSIM [36]. Note that the value of MS–SSIM (M) is $-10\log_{10}\left(1-M\right).$

Table 1 summarizes the rate savings and image quality differences of PE methods and JPEG-compressed images using BD measures for the RD curves presented in Figure 6, Figure 7 and Figure 8. First, we considered the JPEG algorithm for color images compression without chroma subsampling. The results show that the proposed IIB–CPE (8 × 8) preserves the compression savings of color–EtC (16 × 16), while outperforming GS–EtC by almost 15%. However, there is a 3% datarate difference as compared to compression of the original images. The main reason is that the DC coefficients in adjacent blocks have higher correlation and the JPEG algorithm treats them differently to the AC coefficients. The DC coefficients in adjacent blocks are differentially pulse code modulated (DPCM) with each other and their prediction errors are entropy-encoded. In the proposed scheme, the permutation step disrupts this correlation, and compression savings that could have been achieved by the DPCM are lost. On the other hand, for smaller sub-block sizes such as (4 × 4) and (2 × 2), there is an increment in the image size across all methods. However, compared to conventional PE methods, where the file size drastically increases, the proposed method is still able to deliver compression savings. For example, to achieve the highest-quality image (that is, QF=100) when using the smallest block size of (2 × 2), the bitrate of IIB–CPE is 4.6 bpp, whereas for conventional Color–EtC and GS–EtC methods, the bitrate is 10.4 and 8.9 bpp, respectively. The bitrate of conventional PE methods is in fact an increment from the original image size.

Second, we considered the JPEG algorithm for color image compression with a subsampling ratio of 4:2:0. The JPEG encoder performs chroma sub-sampling (such as 4:2:0) on a block size of 16 × 16. Therefore, when the images are encrypted with a block size smaller than the standard specified size, then the resulting (8 × 8) blocks in color components will consist of pixels from different blocks. Such pixels have low correlation and performing interpolation on them to recover the original image resolution results in block artifacts (Type I). In addition, when the luminance component (Y) is shuffled with either of the chrominance components (Cb or Cr) as a result of the channel shuffling step, then during encoding, this block is treated as a chrominance component. Based on human color perception capabilities, the JPEG algorithm processes the chrominance components differently to the luminance. For example, the chrominance component resolution is reduced during the subsampling step and the values are represented with fewer bits during the quantization step. Therefore, when a block that belongs to the luminance component undergoes this processing, then, regardless of its size, the block is recovered with blur distortion (Type II). For visual inspection, Figure 9 shows images recovered from PE encryption methods with different block sizes. For the conventional Color–EtC method, when 16 × 16 block size is used, then the Type II distortion appears in the regions where there is an abrupt change in the pixels values. Images encrypted with block sizes smaller than 16 × 16 have both Type I and Type II distortions. Since, in the GS–EtC method, the chroma subsampling is performed before encryption, no block artifacts are visible. However, for smaller block sizes, there is visible distortion as a result of the quantization step. On the other hand, for the proposed IIB–CPE method, only Type II distortion appears in the similar regions to that of the Color–EtC (16 × 16) method. The proposed method preserved the correlation among the adjacent pixels within a block, as discussed in Section 3.2.1; therefore, it avoids the Type I distortion. The Type II distortion is reduced when using a larger value for the JPEG quality factor, as shown in Figure 10.

For the compression savings when using chroma subsampling, it can be seen in Table 1 that GS–EtC with block size 8 × 8 has achieved better performance than the compared methods. For block size 8 × 8, the method has an improved bitrate compared with that of the plain images. The reason for this is that GS–EtC uses a single quantization table (for example, the standard luminance table in our simulations); therefore, in the low-bitrate region, it has achieved better image quality, as shown in Figure 7. The Color–EtC with block size 16 × 16 has a 6% difference in the bitrate with negligible difference in the image quality. However, for both Color–EtC and GS–EtC, the performance gain is reduced significantly with the smaller block size. On the other hand, the proposed method has a higher bitrate requirement than the conventional methods. However, it is still able to deliver compression savings across different blocks sizes with a negligible difference in image quality. For the proposed method, there is an opposite trend between the measures and the block sizes. This occurs because when using sub-blocks of smaller sizes for intra block processing, then the correlation is better preserved within the block.

As discussed in Section 2, existing PE methods have a prerequisite of color image as an input for better security. As opposed to conventional methods, the encryption efficiency of proposed method is independent of input image representation because of the intra block processing. Therefore, we have also evaluated compression efficiency of existing EtC and IIB–CPE methods on grayscale images. Table 1 summarizes the rate savings and image quality differences of PE methods and JPEG-compressed images using BD measures on grayscale images. Note that since the input are grayscale images with only one channel, Color–EtC and GS–EtC can be implemented with the same algorithm and are together referred to as ‘EtC’ method in Figure 8 and Table 1. The results show that for IIB–CPE with sub-block sizes of 4 × 4 and 2 × 2 has acceptable degradation in compression savings, whereas conventional EtC methods were unable to preserve compression savings for smaller blocks sizes.

### 4.3. Encryption Analysis

Several statistical tests, for example, histogram analysis, correlation analysis, and entropy analysis, usually assess the security of encryption algorithms. These tests are commonly used for analysis of full encryption techniques, where an attacker tries to discover details of the algorithm. However, different from full encryption algorithms, the goal of perceptual encryption cryptanalysis is to recover a better-quality image out of the unencrypted data and their semantics [37]. The compressible PE preserves correlation of the original images on a bock level, which may be vulnerable to the jigsaw puzzle solver attack proposed in [20]. It is a ciphertext-only attack (COA) where the main goal is to reconstruct the original image fully or partially from the cipher image by exploiting the correlation that exist in each block. To demonstrate robustness against the attack, we extended the jigsaw puzzle attack to accommodate sub-block processing. The proposed jigsaw puzzle solver reconstructs the original image in two steps. First, correct orientations of sub-blocks in an entire block are recovered. For this purpose, the cipher image can be treated as a type of jigsaw puzzle where only orientation of the pieces are unknown. Here, the change in orientation is a result of the combination of rotation and inversion transformations, which is given by a composite function as
(7)$$f_{R,I}\left(p_{i}\right)=f_{R}\circ f_{I}\left(p_{i}\right),$$

The function $\;f_{R}\;$ rotates a piece where  R∈{0°,90°,180°,270°}, and the function $f_{I}\;$flips a piece, where  I∈{H, V, HV, No Flip} and H: horizontal flip, V: vertical flip, and HV: horizontal flip followed by vertical flip. For the composite function, the rotation function  R∈{0°,90°}  or  R∈{180°,270°}  in order to avoid collision. The orientation can be recovered by minimizing the total sum of the cost across the boundaries of any two given pieces. The cost is a pairwise compatibility between two pieces calculated as the Mahalanobis Gradient Compatibility (MGC) measure proposed in [23]. For example, for two pieces $\;p_{1}\;$ and $\;p_{2}$, the compatibility between top of$\;\;p_{1}\;$ and bottom of $\;p_{2}\;$ is represented as $\;C_{TB}\left(p_{1},p_{2}\right)$. The minimum compatibility of the two pieces for the proposed solver is given as
(8)$$C_{TB}\left(p_{1},p_{2}\right)=\underset{f_{R,I}}{\min}\left\{C_{TB}\left(p_{1},f_{R,I}\left(p_{2}\right)\right)\right\}.$$

Since the positions of the sub-blocks are not changed, each sub-block is compared only with its neighbor in the block. Once the sub-blocks’ orientation is recovered, the next step is to solve the puzzle for the transformation performed on the entire blocks. Note that the recovery of the sub-blocks’ orientation with respect to their neighbors in a block does not necessarily guarantee the correct orientation of the entire block. Therefore, the entire block has the transformation of $\;f_{RI}\;$ and has an additional function $\;f_{NP}\;$ that applies negative–positive transformation on a block, where  NP∈{0, 1}  and 1 being transformation is applied. The overall transformations on a block $\;P_{i}\;$ can be defined by a composite function as
(9)$$f_{R,I,NP}\left(P_{i}\right)=f_{R}\circ f_{I}\circ f_{NP}\left(P_{i}\right),$$
and the minimum compatibility in Equation (8) for two blocks $\;P_{1}\;$ and $\;P_{2}\;$ becomes:(10)$$C_{TB}\left(P_{1},P_{2}\right)=\underset{f_{R,I,NP}}{\min}\left\{C_{TB}\left(P_{1},f_{R,I,NP}\left(P_{2}\right)\right)\right\}.\;$$

The position of each block is changed because of the permutation step in the encryption; therefore, the compatibility of each block should be computed with every other block in the puzzle. Finally, the calculated compatibility scores are then used to solve the puzzle by using a constrained minimal spanning tree algorithm proposed in [23]. The following measures proposed in [23,38] were investigated in this study to show the robustness against the jigsaw puzzle solver attacks:

Direct comparison (**D_c_**): measures the ratio of blocks in the correct position in the recovered image. Let  I  be the original image, $I_{r}\;$the recovered image, $p_{i}$ the ith piece, and  N  the total number of pieces, then$,\;\;D_{c}\left(I_{r}\right)\;$ is given by
(11)$$\{\begin{array}{c} D_{c}\left(I_{r}\right)=\frac{1}{N}\overset{N}{\underset{i=1}{\sum }}d_{c}\left(p_{i}\right),\;\;\;\;\;\;\;\;\;\;\\ d_{c}\left(p_{i}\right)=\{\begin{array}{c} 1,\\ 0, \end{array}\;\;\;\;\;\begin{array}{c} I_{r}\left(p_{i}\right)=I\left(p_{i}\right),\\ I_{r}\left(p_{i}\right)\neq I\left(p_{i}\right). \end{array} \end{array}$$

Neighbor comparison (**N_c_**)**:** measures the ratio of correctly joined pairwise blocks. For the recovered image $\;I_{r}\;$ with B boundaries among the pieces and $\;b_{i}\;$ being the ith boundary, $\;N_{c}\left(I_{r}\right)\;$ is given by
(12)$$\{\begin{array}{c} N_{c}\left(I_{r}\right)=\frac{1}{B}\overset{B}{\underset{i=1}{\sum }}n_{c}\left(b_{i}\right),\;\;\;\;\;\;\;\;\;\;\;\;\;\;\;\;\;\;\;\;\;\;\;\;\\ n_{c}\left(b_{i}\right)=\{\begin{array}{c} 1,\\ 0, \end{array}\;\;\;\;\;\;\;\;\;\;\begin{array}{c} \mathrm{if}\;b_{i}\;\mathrm{is}\;\mathrm{joined}\;\mathrm{correctly},\\ \mathrm{otherwise}. \end{array} \end{array}$$

Largest component comparison (**L_c_**): measures the ratio of the largest joined blocks that have correct pairwise adjacencies with other blocks in the component. For a recovered image $\;I_{r},\;$the $L_{c}\left(I_{r}\right)\;$ is given by
(13)$$L_{c}\left(I_{r}\right)=\frac{1}{N}\underset{i}{\max}\left\{l_{c}\left(I_{r},i\right)\right\},$$where  N is the number of partially correct assembled regions, i=1, 2,⋯,N, and $\;l_{c}\left(I_{r},i\right)\;$ is number of blocks in the ith assembled area. The scores, $\;D_{c},N_{c},L_{c}\in \left[0,1\right],$ where a larger value indicates a better reconstruction of the cipher image. For the robustness analysis, 20 images were randomly chosen from the Tecnick dataset. First, we encrypted the images and then assembled them using the jigsaw puzzle solver. Table 2 summarizes the average $\;D_{c},N_{c},$ and $L_{c}$ scores of those 20 images. The smaller score value of the proposed method is attributed to intra block processing, which reduces the efficiency of the compatibility measure used by the jigsaw puzzle solver.

In addition, to demonstrate the robustness against brute-force attacks, we analyzed the keyspace of the encryption algorithms. The algorithms consisted of four keys: permutation key $\;K_{1}$, rotation and inversion key $\;K_{2}$, negative–positive transformation key $\;K_{3}$, and color-channel shuffling key $\;K_{4}$. The$\;\;K_{i}$ represented as $K_{i}^{inter}$ and $K_{i}^{intra}$, shows transformation on a block and sub-block level, respectively. For an image  I  with  W×H×C  pixels divided into non-overlapping blocks of the same size  w×h  pixels, the number of blocks  N  in each color channel is given by
(14)$$N=\frac{W}{w}\times \frac{H}{h},$$

For intra block processing, when each block of dimension  w×h  pixels is divided into smaller blocks of size $\;w_{s}\times h_{s}$, the number of sub blocks $\;N_{\mathrm{intra}}\;$ is given by
(15)$$N_{\mathrm{intra}}=\left(\frac{w}{w_{s}}\times \frac{h}{h_{s}}\right)\times N,$$

The keyspace $\;K_{Color-EtC}$ for the conventional Color–EtC scheme based on Equation (14) can be derived as:(16)$$K_{Color-EtC}=K_{1}^{inter}\cdot K_{2}^{inter}\cdot K_{3}^{inter}\cdot K_{4}^{inter}\;=N!\cdot 8^{N}\cdot 2^{N}\cdot 6^{N}.\;\;$$

The keyspace $\;K_{GS-EtC}\;$ for the conventional GS–EtC scheme can be derived similarly to Equation (16) without the last term $\;K_{4}^{inter}\;$ as the methods lack color-channel shuffling step:(17)$$K_{GS-EtC}=K_{1}^{inter}\cdot K_{2}^{inter}\cdot K_{3}^{inter}\;=3N!\cdot 8^{3N}\cdot 2^{3N}.$$

The number of blocks is three times the number of blocks used in the color–EtC scheme. The reason is that the GS–EtC scheme concatenates all the three color channels blocks in a single channel, thereby, it transforms each block independent of its color component. When considering the smallest allowable block sizes such as (16 × 16) in Color–EtC schemes and (8 × 8) in GS–EtC schemes then, the number of blocks in GS–EtC are further increased by a factor of four. The keyspace in Equation (17) becomes
(18)$$K_{GS-EtC}=12N!\cdot 8^{12N}\cdot 2^{12N}$$

The keyspace $\;K_{IIB-CPE}$ for the proposed scheme based on Equations (14) and (15) can be derived as:(19)$$K_{IIB-CPE}=K_{1}^{inter}\cdot K_{2}^{intra}\cdot K_{3}^{inter}\cdot K_{4}^{inter}\;\;\;\;\;\;=N!\cdot \left(8^{N_{intra}}\cdot 8^{N}\right)\cdot 2^{N}\cdot 6^{N}.$$

For inter block processing, both Color–EtC and IIB–CPE methods have the same keyspace. However, the intra block processing increases the key size of the IIB–CPE by a factor of $\;8^{N_{intra}}$. More specifically, when the sub-block size is chosen as $\;\left(w_{s}\times h_{s}\right)\in \left\{\left(8\times 8\right),\left(4\times 4\right),\left(2\times 2\right)\right\},\;$then the increment factor $\;8^{N_{\mathrm{intra}}}\in \left\{8^{4N},8^{16N},8^{64N}\right\}$. On the other hand, the GS–EtC methods represent a color input as pseudo grayscale image; therefore, they have larger keyspace than proposed method. However, color image as an input is a prerequisite for GS–EtC methods. Therefore, when the input is a grayscale image, then the achieved encryption efficiency of GS–EtC methods diminishes and proposed method gains advantage of a larger keyspace. For example, as mentioned in Section 3, for grayscale image encryption, all of the conventional EtC methods (Color–EtC, Extended EtC and GS–EtC methods) can be implemented in the same steps. Therefore, their keyspace $\;K_{Conv-EtC}\;$ can be derived as:(20)$$K_{Conv-EtC}=K_{1}^{inter}\cdot K_{2}^{inter}\cdot K_{3}^{inter}\;=N!\cdot 8^{N}\cdot 2^{N}$$

Since the images are compressed by the JPEG standard in grayscale mode, the smallest block size of (8 × 8) can be used. On the other hand, the keyspace $\;K_{IIB-CPE}\;$ of IIB–CPE for grayscale image encryption can be derived by omitting the last term in Equation (19) as:(21)$$K_{IIB-CPE}=K_{1}^{inter}\cdot K_{2}^{intra}\cdot K_{3}^{inter}\;=N!\cdot \left(8^{N_{intra}}\cdot 8^{N}\right)\cdot 2^{N}\cdot$$

For inter block processing, both conventional EtC and IIB–CPE methods have the same keyspace. However, the intra block processing increases the key size of the IIB–CPE in the similar way as discussed for color image encryption. Therefore, it can be seen that the encryption efficiency of proposed method is independent of input image representation because of the intra block processing.

For the completeness of security analysis, it is necessary to show robustness of the proposed method against known-plaintext (KPA) and chosen-plaintext attack (CPA). Similar to the conventional EtC methods, the proposed scheme is robust against KPA because of the use of different keys for encryption of each image. Since the proposed method is a symmetric encryption scheme and does not need to disclose any information about the key, it can resist CPA.

### 4.4. Application: Privacy-Preserving Deep Learning

In contrast to the centralized paradigm, a decentralized system (e.g., *Federated learning (FL)* [39] *where algorithm is distributed instead of data gathering*) relying on the principle of remote executions and distributed data storage provides an infrastructural approach to security and confidentiality [40]. However, a decentralized system does not fully guarantee privacy. For example, a lack of encryption puts the data and algorithm parameters at risk of being stolen or tampered with, and reconstruction of data from the model weights is also possible [41]. In addition, a limited computational capacity or a small amount of data at a node may affect the quality of the results [41]. In this regard, data manipulation techniques, such as differential privacy (DP) [42] and secure aggregation techniques, such as homomorphic encryption (HE) [43], can provide security in FL. However, DP can degrade the data and may reduce the accuracy of the model, especially in domains with limited data, for example, medical imaging [41]. The main challenge in implementing HE is the computational cost and requirement of a specifically designed algorithm to enable computation in the encryption domain [41]. Similarly, secure multiparty computation (SMPC) [44] has the disadvantage of communication overhead.

As an application of the IIB–CPE scheme, we propose an end-to-end image communication system for privacy-preserving deep learning-based image classification. The proposed system considers a secure and efficient transmission of images to a remote location, thereby, *end-to-end,* and enables classification in the encryption domain without the need for decryption, thereby, *privacy-preserving deep learning.* For classification performance, we used the PyramidNet [45] model of 110 layers in depth and a widening factor of α = 270 with ShakeDrop regularization [46]. The model was trained for 300 epochs using Stochastic Gradient Descent (GSD) with the Nesterov accelerated gradient and momentum method. The momentum of 0.9, weight decay of 0.0001, and batch size of 128 were used during training. The initial learning rates were set to 0.1 for CIFAR10 and 0.5 for CIFAR100 datasets, which were then decayed by a factor of 0.1 at 150 and 225 epochs. Figure 11 summarizes the classification accuracy of PPDL models. The learnable PE algorithms LE and PBE have the closest accuracy to plain images; however, they are vulnerable to chosen-plaintext attacks, as demonstrated by [9]. ELE, the most secure PE algorithm, degrades classification accuracy. Among the learnable PE methods, chaos-based SPBE preserved the classification accuracy of plain images while providing a necessary level of security. However, like any other learnable PE method, they do not provide compression savings. In contrast, for compressible PE algorithms, Color–EtC (16 × 16) has the highest accuracy, but its smaller keyspace and larger block size may compromise system security. However, its variant Color–EtC (4 × 4) and conventional GS–EtC (8 × 8) have improved security, but their classification accuracy is reduced, along with compression savings. For the proposed method, when the sub-block size is 8×8, a comparable classification accuracy can be achieved while preserving the compression savings and improving the security of the conventional EtC schemes. Furthermore, for smaller block sizes, there was a slight reduction in the accuracy of the DL model, but improved security strength.

In Figure 11, the same model and datasets were used for all of the PE methods for fair comparison. In the literature, Color–EtC cipher images were used with two isotropic networks to enable privacy-preserving classification. In their analysis, the best accuracy scores of 87.89% and 92.76% were achieved on the CIFAR 10 dataset for vision transformer (ViT–16) [47] and ConvMixer [48], respectively. For the same encryption method (Color–EtC (16 × 16)), our PPDL model achieved almost 6% and 1% better accuracy than ViT and ConvMixer, respectively. However, the main advantage of using ConvMixer is the smaller number of parameters than in the proposed method. On the other hand, when the images were encrypted with the proposed IIB–CPE method, then the achieved accuracy was up to 5% better than the ViT model and almost 2% less than ConvMixer across different smaller block sizes. As mentioned earlier, the use of larger block size makes the Color–EtC method vulnerable to the COA attack. Therefore, the IIB–CPE based PPDL has the main advantage of better security.

## 5. Discussion

A communication system has dual requirements of compression and encryption in order to provide efficient and secure transmission of data [3]. For an image communication system to enable privacy-preserving computation, it must fulfill the following requirements: (1) bandwidth efficiency and security during data transmission, (2) the images should be encrypted in such a way that the information (for example, color, and perceptual information, etc.) vital to the application should remain intact while providing provable security. Based on these requirements, Table 3 presents a comparison between different perceptual encryption methods. As mentioned previously, images encrypted by the learnable encryption methods are not compressible and among them, only the SPBE method delivered two requirements. For compressible PE algorithms, GS–EtC is the most secure option; however, both compression savings and model accuracy have been suffered. The color–EtC scheme was successful in preserving the file size and model accuracy; however, the method fell short of meeting the encryption requirements. Finally, the proposed IIB–CPE method met all the requirements of an image communication system for PPDL. One limitation of the proposed method is that for color images compression, the savings reduced with smaller block size. However, compared to conventional methods, which were unable to provide any compression savings for smaller block size and in fact increased the file size, IIB–CPE was able to reduce the images sizes. Similarly, when the JPEG algorithm is implemented with chroma subsampling, then the rate difference is increased for the proposed method. The main reason for this is the Type II distortion, which is a result of the color channel shuffling step in the proposed method. One solution for mitigating this problem is to use the same quantization table for both luminance and chroma components as used in GS–EtC methods. In addition, when the classification task becomes more complicated, for example, classifying the CIFAR 100 dataset, the model performance is reduced across all encryption methods. The performance can be improved by utilizing an adaptation network that can learn features in the shuffled images as proposed in [7], but at the cost of additional parameters training. An alternative solution is to consider the proposed PE method with isotropic networks such as ViT and ConvMixer models for an image classification task, as utilized in [19].

## 6. Conclusions

In this study, we propose a compressible perceptual encryption algorithm based on inter and intra block processing, which improved the encryption efficiency of existing methods without compromising the compression savings. The proposed method provided security and bandwidth efficiency during transmission. The JPEG format compatibility of the encoded images makes them suitable for a wide range of applications, such as cloud-based photo storage, privacy-preserving content-based image retrieval, and social networking services. In addition, the main advantage of the proposed method is that it retained color information, which made them suitable for privacy-preserving computations without the need of decryption.

## Figures and Tables

**Figure 1 sensors-22-08074-f001:**
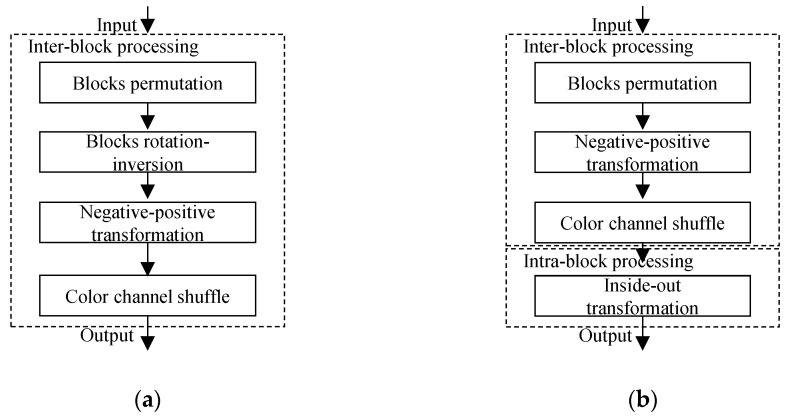
Block-based perceptual encryption algorithms. (**a**) The conventional perceptual encryption scheme. (**b**) The proposed inter and intra block processing-based perceptual encryption scheme.

**Figure 2 sensors-22-08074-f002:**
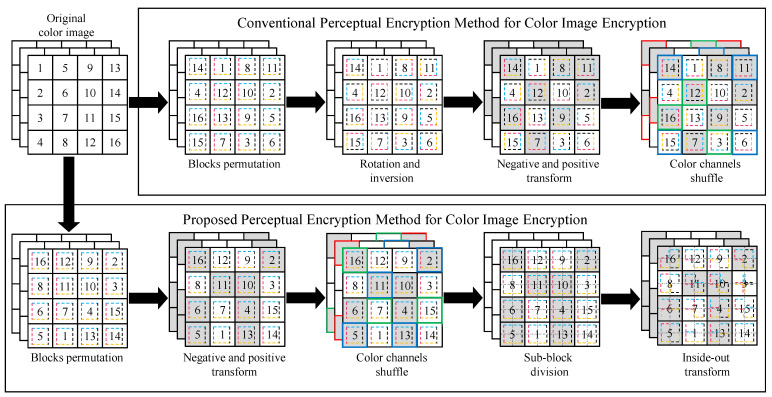
Comparison of existing and proposed perceptual encryption methods for color image encryption. The label of a block shows its original location while the dashed square inside a block shows its orientation. The negative–positive transformed blocks are the shaded ones. The color of a block border shows shuffled channels. The dotted line shows sub-block division.

**Figure 3 sensors-22-08074-f003:**
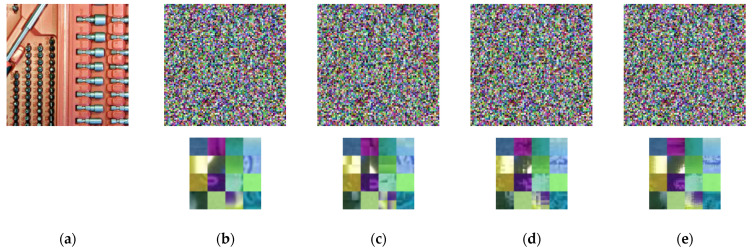
Visual analysis of inter and intra block processing on example image from Tecnick dataset. (**a**) Plain image. (**b**) Conventional EtC method (16 × 16). (**c**–**e**) Encrypted image of the proposed method with sub–block size (8 × 8), (4 × 4), and (2 × 2), respectively. The last row shows enlarged image of the bottom left corner in each encrypted image. Compared to conventional methods, the proposed method achieves visual encryption of local details.

**Figure 4 sensors-22-08074-f004:**
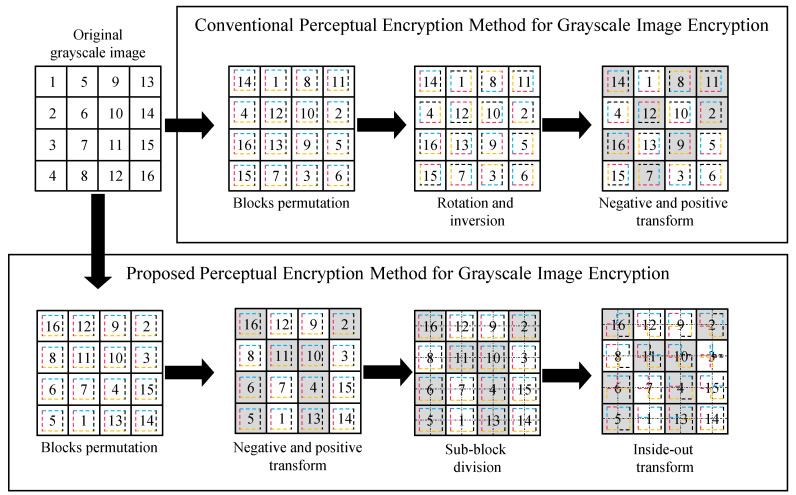
Comparison of existing and proposed perceptual encryption methods for grayscale image encryption. The label of a block shows its original location while the dashed square inside a block shows its orientation. The negative–positive-transformed blocks are the shaded ones. The dotted line shows sub-block division.

**Figure 5 sensors-22-08074-f005:**
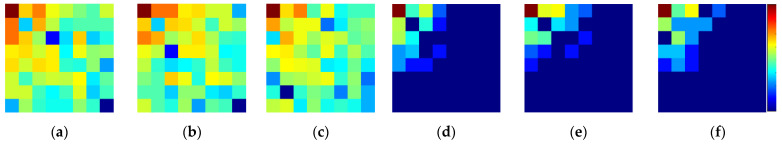
The DCT transform of the original and scrambled images. (**a**–**c**) DCT coefficients and (**d**–**f**) their quantized representations with $\;Q_{f}=80$. (**a**) Original image. (**b**) Conventional EtC image. (**c**) IIB–CPE image.

**Figure 6 sensors-22-08074-f006:**
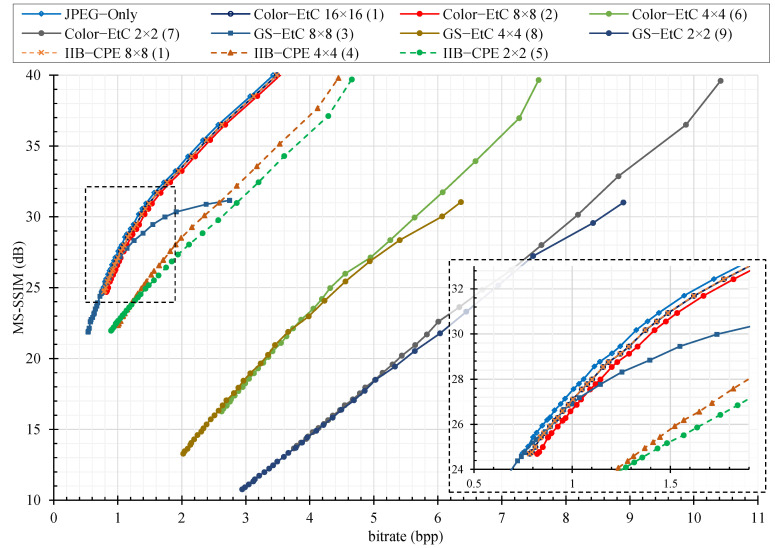
The JPEG compression performance without chroma subsampling on different perceptual encryption methods in terms of rate distortion (RD) curves with respect to MS–SSIM (dB) on Tecnick color dataset. The number enclosed in parentheses at the end of each series name shows its performance rank. The overlapping regions in the graph are zoomed in and shown in the bottom right corner.

**Figure 7 sensors-22-08074-f007:**
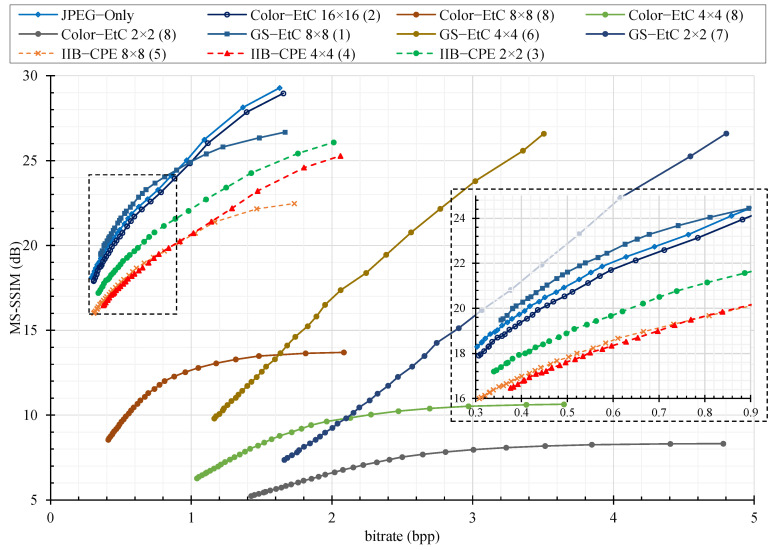
The JPEG compression performance with chroma subsampling ratio (4:2:0) on different perceptual encryption methods in terms of rate distortion (RD) curves with respect to MS–SSIM (dB) on Tecnick color dataset. The number enclosed in parentheses at the end of each series name shows its performance rank. The overlapping regions in the graph are zoomed in and shown in the bottom right corner.

**Figure 8 sensors-22-08074-f008:**
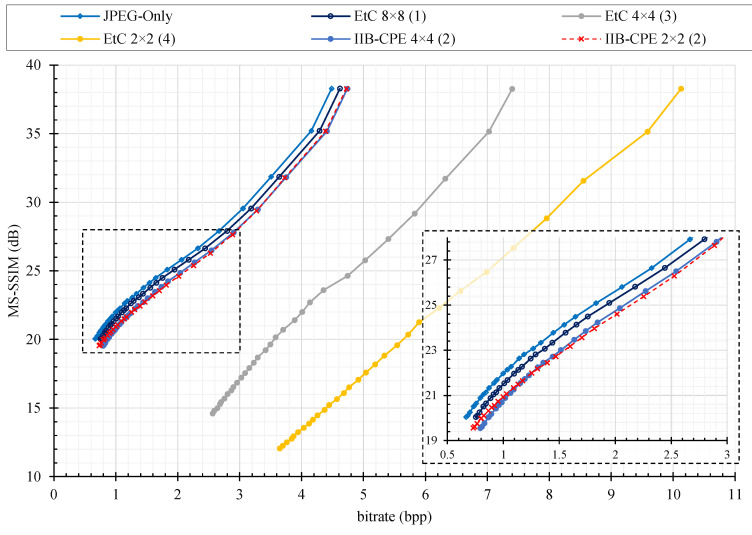
The JPEG compression performance on different perceptual encryption methods in terms of rate distortion (RD) curves with respect to MS–SSIM (dB) on Shenzhen grayscale images dataset. The number enclosed in parentheses at the end of each series name shows its performance rank. The overlapping regions in the graph are zoomed in and shown in the bottom right corner.

**Figure 9 sensors-22-08074-f009:**
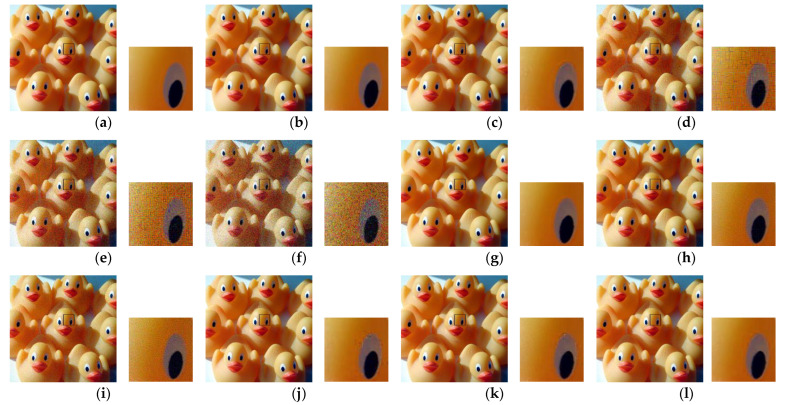
Images recovered from PE methods using different block sizes. The JPEG quality factor is 71. (**a**) The original image. (**b**) Image recovered from the plain image. (**c**–**f**) Images recovered from the Color–EtC method with block sizes (16 × 16, 8 × 8, 4 × 4, and 2 × 2), respectively. (**g**–**i**) Images recovered from the GS–EtC method with block sizes (8 × 8, 4 × 4, and 2 × 2), respectively. (**j**–**l**) Images recovered from the proposed with block sizes (8 × 8, 4 × 4, and 2 × 2), respectively. The boxed region in each image is zoomed in and shown on its left side.

**Figure 10 sensors-22-08074-f010:**
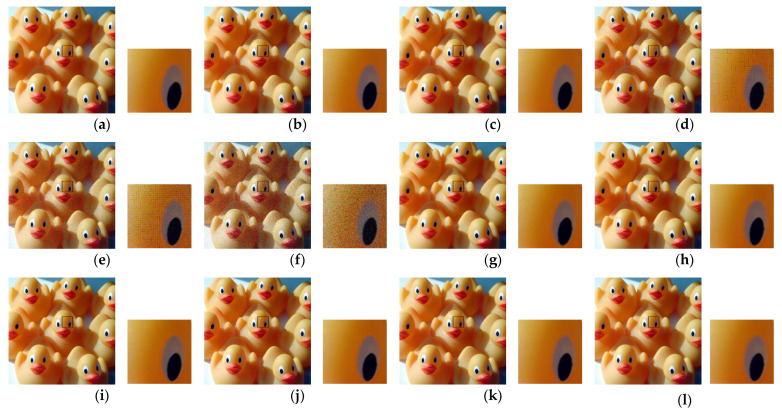
Images recovered from PE methods using different block sizes. The JPEG quality factor is 100. (**a**) The original image. (**b**) Image recovered from the plain image. (**c**–**f**) Images recovered from the Color–EtC method with block sizes (16 × 16, 8 × 8, 4 × 4, and 2 × 2), respectively. (**g**–**i**) Images recovered from the GS–EtC method with block sizes (8 × 8, 4 × 4, and 2 × 2), respectively. (**j**–**l**) Images recovered from the proposed with block sizes (8 × 8, 4 × 4, and 2 × 2), respectively. The boxed region in each image is zoomed in and shown on its left side.

**Figure 11 sensors-22-08074-f011:**
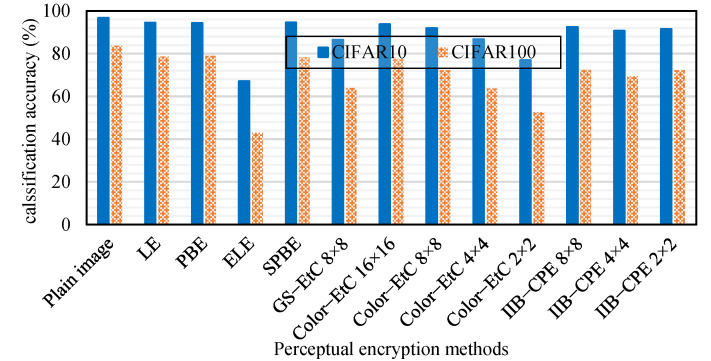
Classification accuracy (%) of the deep learning model in the encryption domain on CIFAR10 and CIFAR100 datasets.

**Table 1 sensors-22-08074-t001:** The JPEG compression performance on different perceptual encryption methods in terms of Bjøndegaard Delta measures. The rate differences are for the equivalent quality relative to the JPEG under MS–SSIM for RD curves plotted in Figure 6, Figure 7 and Figure 8.

Input Image	Sub-Sampling Ratio	Methods	Block-Size	BD-Rate	BD-MS-SSIM
Color	4:4:4	Color–EtC	16 × 16	3.18	−0.32
			8 × 8	6.18	−0.61
			4 × 4	267.72	−19.88
			2 × 2	424.69	−27.8
		GS–EtC	8 × 8	17.76	−2.16
			4 × 4	405.32	−20.04
			2 × 2	646.57	−27.39
		IIB–CPE	8 × 8	3.11	−0.31
			4 × 4	64.04	−5.4
			2 × 2	78.61	−6.01
	4:2:0	Color–EtC	16 × 16	5.98	−0.39
			8 × 8	nan	−12.55
			4 × 4	nan	−19.94
			2 × 2	nan	−23.39
		GS–EtC	8 × 8	−1.41	−0.21
			4 × 4	352.1	−16.33
			2 × 2	549.1	−22.26
		IIB–CPE	8 × 8	112.44	−4.1
			4 × 4	105.81	−4.19
			2 × 2	60.52	−2.76
Grayscale	–	EtC	8 × 8	5.36	−0.51
			4 × 4	126.67	−12.77
			2 × 2	220.15	−21.45
		IIB–CPE	4 × 4	11.92	−1.12
			2×2	11.82	−1.13

**Table 2 sensors-22-08074-t002:** Robustness of the perceptual encryption methods against jigsaw puzzle solver attacks. (**N_c_**: neighbor comparison, **L_c_**: largest component comparison, **D_c_**: direct comparison).

Methods	N_c_	L_c_	D_c_
Color–EtC 16 × 16	0.11	0.12	0.01
GS–EtC 8 × 8	0.001	0.002	0.001
IIB–CPE 8 × 8	0.08	0.02	0.01
IIB–CPE 4 × 4	0.05	0.02	0.01
IIB–CPE 2 × 2	0.06	0.02	0.01

**Table 3 sensors-22-08074-t003:** Performance comparison of different perceptual encryption methods to enable privacy-preserving deep learning.

Methods	Compression	Encryption	PPDL
LE	×	◯	✓
PBE	×	◯	✓
ELE	×	✓	◯
SPBE	×	✓	✓
GS–EtC	◯	✓	◯
Color–EtC	✓	◯	✓
IIB–CPE	✓	✓	✓

✓: can enable a task with desired efficiency, ◯: can enable a task but the efficiency suffers, and ×: cannot enable a task.

## Data Availability

All the datasets used in this study are publically available. The CIFAR datasets used for privacy-preserving deep learning analysis is accessible at: https://www.cs.toronto.edu/~kriz/cifar.html (accessed on 16 December 2021). The Tecnick dataset used for color image compression and encryption is accessible at: https://testimages.org/ (accessed on 16 December 2021). The Shenzhen dataset used for grayscale image compression analysis is accessible at: https://ceb.nlm.nih.gov/repositories/tuberculosis-chest-X-ray-image-data-sets/31 (accessed on 13 March 2022).

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
