# Peer review of "IIB–CPE: Inter and Intra Block Processing-Based Compressible Perceptual Encryption Method for Privacy-Preserving Deep Learning†"

_sensors, 2022, doi:10.3390/s22208074_

Round 1

Reviewer 1 Report

In this paper, an improved Encryption-then-Compression (EtC) scheme for JPEG images is proposed, and then is extended to privacy-preserving deep learning application. Some of the issues are listed as below:

1. Simplify the title as "IIB–CPE: A Compressible Perceptual Encryption Method for Privacy–Preserving Deep Learning"

2. At line 147, it is stated that the proposed method can solve the limitation that the input should be color images. However, it seems that the proposed method still requires a color image as input (see Fig.2 in the paper).

3. In Fig.5a), most of the important curves are clustered in the upper left corner. Zoom this area to show the gap between different schemes.

4. The chroma subsampling, such as 4:2:0 sampling, can cause image distortion as shown in reference [1]. Does this problem  also exist in the proposed encryption scheme ? Discuss how to address the problem if it exists, or how to avoid it.

5. In the analysis of key space, it would be better if  the block size was taken into account.

6. For the completeness of security analysis, the robustness of the proposed encryption scheme against chosen-/known-plaintext attacks should also be discussed.

7. The color EtC scheme has already introduced in privacy-preserving image classification task in reference [2]. Discuss the distinction and advantages of the proposed method.

[1] T. Chuman, W. Sirichotedumrong, and H. Kiya, “Encryption-Then-Compression Systems Using Grayscale-Based Image Encryption for JPEG Images,” *IEEE Transactions on Information Forensics and Security*, vol. 14, no. 6, pp. 1515–1525, Jun. 2019, doi: [10.1109/TIFS.2018.2881677](https://doi.org/10.1109/TIFS.2018.2881677).

[2] A. Maungmaung and H. Kiya, “Privacy-Preserving Image Classification Using Isotropic Network,” *IEEE MultiMedia*, pp. 1–1, 2022, doi: [10.1109/MMUL.2022.3168441](https://doi.org/10.1109/MMUL.2022.3168441).

8. As for insecurity of traditional chaos theory-based full encryption algorithms, refer to https://dx.doi.org/10.1016/j.jvcir.2021.103424

9. Re-write the conclusion part into ONE paragraph with the concluding tense:

http://cc.oulu.fi/~smac/TRW/tense_conclusions.htm

http://writingcenter.unc.edu/handouts/conclusions

Reviewer 2 Report

The manuscript “IIB–CPE: A Compressible Perceptual Encryption Method based 2 on Inter and Intra Block Processing for Privacy–Preserving 3 Deep Learning”

An inter and intra block processing for compressible PE methods (IIB–CPE) method was proposed. The method represents an input as a color image and performs block level inter processing and sub–block level intra processing on it.

This manuscript is well organized, however there are several questions, and the manuscript should have minor revision.  

1.     Have you choose real images to validate your proposed method?

2.     Does the method have any limitations? Please discuss it in your manuscript.

Reviewer 3 Report

Manuscript No.: Sensors-1935005

IIB–CPE: A compressible perceptual encryption method based on inter and intra block processing for privacy–preserving deep learning

Ijaz Ahmad and Seokjoo Shin

The manuscript under review proposes inter and intra block processing for compressible perceptual encryption. A color image has been used as input and block level inter processing and sub–block level intra processing have been performed. It has been claimed that proposed method performs better as compared to existing perceptual encryption methods.

Comments:

1.  As much as possible abbreviations should not be used in the title of a manuscript. Therefore, here the term ‘IIB-CPE’ should be expanded.

2.    Optical image encryption-decryption is an established alternative technique for information security. There compression of encrypted data has been studied extensively. Authors should look into the literatures on the topic such as Adv. Opt. Photon. 1 (2009) 589-636. Also, in 2019, IOP Publs published a book titled Optical Cryptosystems and the references cited therein.  

3.     Mathematical analysis of the proposed encryption scheme is missing.
